# Characterization and Correction of OMPS Nadir Mapper Measurements for Ozone Profile Retrievals

Juseon Bak[a,#] (juseon.bak@cfa.harvard.edu), Xiong Liu[b](xliu@cfa.harvard.edu),

Jae-Hwan Kim[a, *] (jaekim@pusan.ac.kr), David P. Haffner[c] (david.haffner@ssaihq.com),

Kelly Chance[b] (kchance@cfa.harvard.edu), Kai Yang[d](KaiYang@umd.edu),

Kang Sun[b](Kang.sun@cfa.harvard.edu)

[a]*Pusan National University, Busan, Korea*

[b]*Harvard-Smithsonian Center for Astrophysics, Cambridge, MA, United States*

[c]*Science Systems and Applications, Inc., 10210 Greenbelt Rd, Lanham, MD 20706, United States*

[d]*Department of Atmospheric and Oceanic Science, University of Maryland College Park, College Park, Maryland, USA*

#Currently at Harvard-Smithsonian Center for Astrophysics, Cambridge, MA, United States

*Corresponding Author

Abstract

This paper verifies and corrects the Ozone Mapping and Profiler Suite (OMPS) Nadir Mapper (NM) Level 1B v2.0 measurements with the aim of producing accurate ozone profile retrievals using an optimal estimation based inversion method to fit measurements in the spectral range 302.5-340 nm. The evaluation of available slit functions demonstrates that preflight-measured slit functions well represent OMPS measurements compared to derived Gaussian slit functions. Our initial OMPS fitting residuals contain significant wavelength and cross-track dependent biases, resulting in serious cross-track striping errors in the tropospheric ozone retrievals. To eliminate the systematic component of the fitting residuals, we apply "soft calibration" to OMPS radiances. With the soft calibration the amplitude of fitting residuals decreases from ~1 % to 0.2 % over low/mid latitudes, and thereby the consistency of tropospheric ozone retrievals between OMPS and the Ozone Monitoring Instrument (OMI) is substantially improved. A common mode correction is also implemented for additional radiometric calibration; it improves retrievals especially at high latitudes where the amplitude of fitting residuals decreases by a factor of ~2. We estimate the noise floor error of OMPS measurements from standard

deviations of the fitting residuals. The derived error in the Huggins band (~0.1 %) is twice the OMPS
L1B measurement error. OMPS noise floor errors better constrains our retrievals, leading to improving
information content of ozone and reducing fitting residuals. The final precision of the fitting residuals
is less than 0.1 % in the low/mid latitude, with ~1 degrees of freedom for signal for the tropospheric
ozone, meeting the general requirements for successful tropospheric ozone retrievals.

## 1. Introduction

Atmospheric ozone has very different roles depending upon its altitude. About 90 % of the total
ozone is in the stratosphere, protecting the Earth's life from harmful solar ultraviolet (UV) radiation
that can cause skin cancer and immune system suppression. The remaining 10 % in the troposphere
shows dangerous effects as a major component of photochemical smog at surface level and as a short-
lived greenhouse gas in the upper troposphere, whereas in the middle troposphere it plays a beneficial
role in chemically cleaning the atmosphere as a precursor of hydroxyl radicals (OH). Therefore, vertical
ozone profiles should be monitored to improve our understandings of the chemical and physical
functions of this important trace gas. Space-based monitoring of ozone profiles including the
troposphere from backscattered UV radiation has been available since the launch of Global Ozone
Monitoring Experiment (GOME) (European Space Agency, 1995) on board the Second European
Remote Sensing Satellite (ERS-2) in April 1995. Its successors continued the role of GOME for
atmospheric ozone monitoring with Scanning Imaging Absorption SpectroMeter for Atmospheric
CHartographY (SCIAMACHY) (Bovensmann et al., 1999) aboard the Environmental Satellite
(ENVISAT), GOME-2s (EUMETSAT, 2006) aboard the MetOp-A and MetOp-B, and Ozone
Monitoring Instrument (OMI) (Levelt et al, 2006) flown on the EOS Aura spacecraft. The good
performance of OMI ozone profile retrievals in both stratosphere and troposphere has been
demonstrated through extensive validation efforts using ozonesondes, aircraft, satellite data, and
ground-based total ozone data (Pittman et al., 2009; Liu et al., 2010b; Bak et al., 2013b; 2015; Huang
et al., 2017a, b). However, a portion of OMI radiance measurements has been affected by the partial
blockage of the instrument's entrance slit, a problem termed the row anomaly, which started in 2007
and grew serious in January 2009 (Schenkeveld, 2017). The Ozone Mapping and Profiler Suite (OMPS)
aboard the Suomi National Polar-Orbiting Partnership (NPP) satellite launched in 2011 (Flynn, et al.,
2014) represents the next generation of US instruments to continue the role of OMI in monitoring total
ozone and ozone vertical profiles, together with the TROPOspheric Monitoring Instrument (TROPOMI)
to be launched on board the Sentinel-5 Precursor satellite in 2017 (Veefkind et al., 2012). OMPS is a
sensor suite which consists of three instruments, the Nadir Mapper (OMPS-NM), the Nadir Profiler
(OMPS-NP), and the Limb Profiler (OMPS-LP). The OMPS-NM is designed to measure the daily
global distribution of total column ozone with an $110^o$ cross-track field of view (FOV), similar to OMI
and the Total Ozone Monitoring Spectrometer (TOMS) series (Bhartia and Wellemeyer, 2002). OMPS-
NP is an ozone profiler sensor, measuring the vertical ozone profiles in the upper stratosphere, similar
to the Solar Backscatter Ultraviolet (SBUV/2) series (Bhartia et al., 2013). The OMPS-LP is designed
to measure ozone profiles in the stratosphere and upper troposphere at high vertical resolution, similar
to the Microwave Limb Sounder (MLS). Both OMPS-NP and OMPS-LP are ozone profile sensors, but
lack sensitivity to the troposphere due to the spectral coverage of 250-290 nm and the viewing geometry,
respectively. Therefore, OMPS-NM is the only candidate for global monitoring of ozone profiles down
to the troposphere even though its spectral resolution of 1.0 nm does not fully resolve the ozone
absorption band features in the Huggins band and its spectral coverage of 300-380 nm is insufficient to
retrieve stratospheric ozone profiles. The retrieving of ozone profiles including tropospheric ozone from
OMPS-NM measurements has not yet been presented in the literature. The present effort fills the gap
between OMI and upcoming satellite observations.
The final goal of this study is to demonstrate the successful performance of ozone profiles and
tropospheric ozone retrievals from only OMPS-NM measurements. Thus, we refer to OMPS-NM
simply as OMPS hereafter. The retrieval algorithm used in this study is based on the Smithsonian
Astrophysical Observatory (SAO) ozone profile algorithm that was developed for GOME (Liu et al.,
2005) and OMI (Liu et al., 2010a). The SAO OMI algorithm is based on an optimal estimation inversion
(Rodgers, 2000) combined with accurate wavelength/radiometric calibration, forward model simulation,
and good a priori knowledge. This algorithm has been implemented for ozone profile and $SO_2$ retrievals
from GOME-2 instrument (Cai et al., 2011; Nowlan et al., 2011) and will be adapted to ozone profile
retrievals from upcoming geostationary UV/VIS spectrometers including the Geostationary
Environmental Monitoring Spectrometer (GEMS) (Bak et al 2013a) and Tropospheric Emissions:
Monitoring of POllution (TEMPO) instrument (Chance et al., 2013, Zoogman et al., 2017) for
monitoring air quality over North America and East Asia, respectively. OMPS has a similar instrument
concept to OMI, GEMS, and TEMPO and hence the application of the similar retrieval algorithms to
these measurements will provide an excellent opportunity for long-term trend analysis of ozone profiles,
especially in the troposphere. The OMI algorithm is very similar to our OMPS algorithm, but it needs
additional optimization for OMPS. In this paper we focus largely on characterizing OMPS
measurements (1) through the cross-correlation between OMPS irradiances and a high-resolution solar
reference to be used in the verification of OMPS slit function measurements and the characterization of
the wavelength registration and (2) through extracting the systematic and random components of fitting
residuals between measured and calculated normalized radiances to be used in radiometric and
measurement error calibrations, respectively. Several companion papers to follow will deal with the
detailed error analysis, retrieval characteristics of the retrieved ozone profiles, and validation of
retrievals.
The paper is divided into four sections: First, we give a description of OMPS-NM Level 1B (L1B)
v2.0 data (Jaross, 2017) and the ozone profile algorithm in Sect. 2. Section 3 discusses the
wavelength/slit function calibrations and measurement corrections for radiance and measurement error,
respectively. Conclusions are in Sect. 4.

## 2. Data and Method

## 2.1 OMPS measurements

The Suomi NPP satellite is a NOAA/NASA scientific partnership, launched in 2011 into a 824 km sun-
synchronous polar orbit with ascending node equator-crossing time at 13:30 local time. Routine
operations began in 2012. Suomi NPP carries five instruments: The Visible/Infrared Imager Radiometer
Suite (VIIRS), the Cross-track Infrared Sounder (CrIS), the Advanced Technology Microwave Sounder
(ATMS), the Ozone Mapping and Profile Suite (OMPS), and the Clouds and the Earth Radiant Energy
System (CERES). OMPS is a key instrument on Suomi NPP. The sensor suite has both nadir and limb
modules. The nadir module combines two sensors: The Nadir Mapper for measuring total column ozone,
and the Nadir Profiler for ozone vertical profile. The Limb Profiler module is designed to measure
vertical ozone profiles with high vertical resolution from the upper troposphere/lower stratosphere to
the mesosphere. The OMPS-NM employs a 2-D CCD that samples spectrally in one dimension and
spatially in the other, similar to OMI. It has a $110\,^{\circ}$ cross-track field of view, resulting in 2800 km
instantaneous swath coverage at the earth's surface; this is sufficient to provide daily global coverage.
It makes 400 swath lines per orbit with 36 cross-track measurements per swath line, resulting in a nadir
footprint of 50 km × 50 km in its nominal configuration. Note that OMPS L1B data used in this
investigation contain 36 cross-track pixels, because the L1B processing in the NASA Ozone SIPS
retains the two central (near-nadir) instantaneous fields of views (IFOVs, 30 km × 50 km and 20 km ×
50 km), without aggregating them into the nominal 50 km × 50 km pixel. The spectral coverage is from
300 to 380 nm with a spectral resolution of ~ 1.0 nm and a sampling of 0.42 nm. The OMPS level 0 to
1b processor was recently updated from version 1.0 to 2.0. The satellite measurements from the OMPS-
NM instrument used in this study are from version 2 of the NMEV-L1B data product (Jaross, 2017)
available from the NASA Goddard Earth Sciences Data and Information Services Center (GES
DISC). The data consist of calibrated Earth-view radiance and solar irradiance data measured by the
instrument between 300-380 nm. Seftor et al. (2014) documented many aspects of the previous version
of the dataset that remain the same, but a number of changes for the V2 dataset do reflect advances in
the characterization of the NM sensor (Seftor and Jaross, 2017) which are relevant to this study. These
are summarized as follows: 1) recalculation of instrument band-pass functions in the 300-310 nm region
affected by the dichroic element of the nadir instrument, 2) improved wavelength registration, 3) an
update to the instrument radiance calibration, and 4) improvement to the stray light correction. The
wavelengths below 302 nm are not used in this study, according to the recommendation of the OMPS
science team.

## 2.2 OMPS simulations

We use the Vector LInearized Discrete Ordinate Radiative Transfer (VLIDORT) model (Spurr, 2006;
2008) to simulate OMPS radiances. VLIDORT is also able to simulate the analytic derivatives of
radiance with respect to any atmospheric or surface parameter due to its full linearization capability.
The polarization of light is taken into account in VLIDORT calculation, but the Ring spectrum is
modeled using a single scattering RRS model (Sioris and Evans, 2000). We consider only Rayleigh
scattering (no aerosol) and ozone absorption (no other trace gases), with Lambertian reflectance
assumed for the surface and for clouds. Clouds are treated as a Lambertian reflector at cloud top, with
a fixed albedo of 0.8 unless it is fully cloudy so that the cloud albedo (>0.80) can be derived. Cloud
fraction is required to simulate partial clouds as the weighted average between clear and cloudy scenes
using the Independent Pixel Approximation (IPA). The forward model inputs used in VLIDORT are
listed in Table 1.

## 2.3 OMPS ozone profile retrievals

The inversion from Backscattered UV measurements to the state of the atmosphere is performed
using the well-known optimal estimation method (Rodgers, 2000). It calculates the a posteriori solution
by iteratively and simultaneously minimizing the cost function consisting of the sum of the squared
differences between measured and simulated radiances and between retrieved and a priori state vectors,
constrained by measurement error covariance matrix and a priori error covariance matrix. The a
posteriori solution and cost function can be written:
$$X_{i+1} = X_i + (K_i^T S_y^{-1} K_i + S_a^{-1})^{-1}[K_i^T S_y^{-1}(Y - R(X_i)) - S_a^{-1}(X_i - X_a)] \qquad (1)$$

$$\chi^2 = \left\| S_y^{-\frac{1}{2}}\{K_i(X_{i+1} - X_i) - [Y - R(X_i)]\}\ )\right\|_2^2 + \left\| S_a^{-\frac{1}{2}}(X_{i+1} - X_a)\ \right\|_2^2. \qquad (2)$$

The inputs to the optimal estimation are defined as follows. $\mathbf{X}$ is the state vector to be retrieved,
consisting of ozone profiles as well as other geophysical parameters and spectroscopic parameters
affecting the observed radiances and hence the retrieval of ozone profile. The 24 partial columns of
ozone in DU are retrieved at 25 pressure levels that are initially set to be $P_i = 2^{-i/2}$ atm for $i =$
$0, 1, \ldots 23$ (1 atm = 1013.25 hPa) with the top of the atmosphere at 0.087 hPa for $P_{24}$. The geophysical
parameters include effective surface albedo and cloud fraction. The calibration parameters consists of
two wavelength shift parameters between radiances and irradiances and between radiances and ozone
cross sections and two scaling parameters for the Ring effect that account for filling-in of Fraunhofer
lines in the solar spectrum due to rotational Raman scattering and mean fitting residuals that may not
be accounted for properly in radiometric calibration. The a priori data for ozone is one of the key optimal
estimation inputs because the retrieval solution comes mainly from a priori information rather than
measurement information where the instrument sensitivity to the true ozone profile is insufficient. The
a priori value ($X_a$) and a priori error covariance ($S_a$) of ozone is taken from the tropopause-based ozone
profile climatology that is optimized to represent the dynamical ozone variability in the upper
troposphere and lower stratosphere (Bak et al., 2013b). The measurement vector $Y$ is defined as the
logarithm of the earthshine radiances normalized to the daily solar irradiance. $S_y$ is a measurement
error covariance matrix that is assumed to be a diagonal matrix with diagonal elements being the squares
of the assumed measurement errors. We use OMI noise floor errors (0.4 % below 310 nm, 0.2 % above,
Huang et al., 2017a) as our preliminary measurement constraint and then derive OMPS noise floor
errors specified in Section 3.4. $R(X)$ is the calculated radiances corresponding to $X$. $K$ is a weighting
function matrix representing partial derivatives of the forward model with respect to the atmospheric
parameters, $K_{ij} = \partial R_i(X)/\partial X_j$. More detailed descriptions can be detailed in Liu et al. (2010a).

## 3. Results

## 3.1 Slit Function and Wavelength Calibration`

It is essential to investigate the best knowledge of the instrument slit function to convolve a high-resolution solar reference spectrum for wavelength calibration as well as to convolve high-resolution trace gas cross sections for simulation of earthshine spectra. A triangular bandpass with a fixed bandwidth of 1.1 nm has been typically used for Total Ozone Monitoring Instrument (TOMS), SBUV, and SBUV/2 monochromators. Slit functions of spectrometers such as OMI and GOME1/2 have been measured prior to launch using a tunable laser or analytically derived assuming a Gaussian-type shape if measured slit functions are unavailable or inaccurate. The OMPS preflight slit functions were characterized for each CCD pixels (196 band centers and 36 cross-track positions), which has been adopted and modified for OMPS trace-gas retrievals such as in Yang et al. (2013; 2014) and Gonzalez Abad et al. (2016). The slit function modification is accomplished in the previous works (Yang et al., 2013, 2014) by stretching and shrinking the slit widths, i.e., by applying a wavelength-dependent scaling factor to the OMPS measured slit functions. According to Yang et al. (2013; 2014), we fit the scaling factor as a slit parameter so that variations in measured slit functions before and after launch could be taken into account.

Figure 1a shows an example of measured OMPS slit functions at 320 nm, illustrating that their shapes seem to be Gaussian and vary considerably over cross-track pixels, especially near the wings. Note that the 36 cross-track positions are denoted from 1 at the left edge and 36 at the right edge. The slit function shapes at $17^{th}$ cross-track position are nearly consistent over wavelengths that we are focusing on for ozone retrievals (Fig. 1.b). Figure 1c displays the full width at half maximum (FWHM) including dependencies in both dimensions of the detector arrays. The spectral variation of the slit widths is insignificant (FWHMs vary by less than 0.01 nm), whereas average slit widths vary significantly across track by over 0.1 nm. This characteristic of measurement slit functions confirms that we should consider their cross-track dependence for OMPS slit functions, but their wavelength dependence is ignorable so that we can avoid the time-consuming convolution process.

We evaluate the usefulness of these measured slit functions for fitting both OMPS radiance and irradiance against the analytical slit functions assusing both standard Gaussian and super Gaussian distributions. We note all the Gaussian shapes used in this analysis are assumed to be symmetric. The Gaussian slit function is expressed as

$$S(\lambda) = \frac{k}{2w\Gamma\left(\frac{1}{k}\right)} exp\left[-\left|\frac{\Delta\lambda}{w}\right|^k\right], \qquad (3)$$

where $k$ is the shape factor and $w$ is the slit width, with relative wavelength to band center wavelength, $\Delta\lambda$. This function can describe a wide variety of shapes just by varying $k$; for $k=2$ it becomes the standard Gaussian and $w$ represents the half width at 1/e intensity (FWHM = $2\sqrt{\ln 2}$ $w$). Compared to the standard Gaussian, the super Gaussian has broader peaks at the top and thinner wings if $k$ is larger than 2 whereas it has sharper peaks and longer tails if $k$ is smaller than 2. $w$ of the super Gaussian function represents the half-width at $1/e^{th}$ intensity (FWHM = $2\sqrt[k]{\ln 2}$). The symmetric or asymmetric standard Gaussian has been commonly assumed to derive OMI, GOME, and GOME-2 slit functions (Liu et al., 2005;2010; Nowlan et al., 2011; Cai et al., 2012; Munro et al., 2016). Recently the hybrid combination of standard and flat-top Gaussian functions has been implemented for characterizing OMI laboratory measurements of slit functions (Dirksen et al., 2006) and deriving airborne instrument slit functions (Liu et al., 2015a;2015b; Nowlan et al., 2016). The concept of this hybrid Gaussian function is very similar to the super Gaussian, but is a rather complex with more slit parameters. The super Gaussian function was introduced and tested as an analytical slit function by Beirle et al. (2017) and Sun et al. (2017a;b).

In general, when accurate measurements of slit functions are not available, the instrument line shape of satellite observation is typically assumed to be the same for both radiance and irradiance measurements, and then can be better determined from irradiances due to lack of atmospheric interference.. We simultaneously and iteratively determine the wavelength and slit calibration parameters through cross-correlation of the measured OMPS irradiances to simulated solar irradiances from a well calibrated, high-resolution solar irradiance reference spectrum (Chance and Kurucz, 2010). The simulation of solar irradiance, $I_s$ is described as

$$I_s(\lambda) = AI_o(\lambda + \Delta\lambda) \times \sum_{i=0}^{2} P_i(\lambda - \lambda_{avg})^i, \quad (4)$$

where $I_o$ is the convolved high-resolution solar reference spectrum with assumed slit functions, A is the scaling parameter for $I_o$. $\lambda + \Delta\lambda$ Indicates the process of wavelength calibration (e.g. shift and squeeze); only the wavelength shift is considered in this study. $P_i$ represents the coefficients of a scaling polynomial (third order in this study). This approach was firstly introduced by Caspar and Chance (1997), and is widely used for wavelength and slit function calibrations in trace gas retrievals from UV/visible measurements.

In this experiment, the slit parameters, $w$ and $k$ or slit scaling are fitted from daily measured
OMPS irradiances over the wavelength range 302-340 nm at each cross-track position. Note that this
slit calibration ignores the wavelength dependence for deriving analytic slit functions and slit scaling to
the measured slit functions; this is a good approximation based on Fig. 1b as the wavelength dependence
of the slit functions is small. But the variation of the slit shape with wavelength could be considered
with OMPS preflight measured slit functions given for every CCD dimension if it becomes necessary.
The left panels of Fig. 2 compare the derived slit parameters from OMPS irradiances using different
functions. The red line of Fig. 2.a.1 shows that a slight change of the preflight-measured slit functions
is required to model the OMPS irradiance measurements, by up to 4% at both edges. Therefore the
benefit of fitting measured slit functions over fixing them is found to be trivial (~ 0.001 %) at nadir
cross-track pixels (12-30[th]); for edge pixels, the improvement in fitting residuals is more noticeable, up
to 0.18%. The shape factor ($k$) of the derived super Gaussian functions is found to be ~ 2.3 for left swath
and ~ 2.5 for right swath (Fig. 2.b.1), implying that they have broader peaks and thinner wings compared
to the standard Gaussian if slit widths are equal. The slit widths of three different slit functions show
similar variations with respect to cross-track positions. The FWHMs vary from widest at ~12[th] cross-
track position to narrowest at the edges, but they are significantly narrower at the rightmost cross-track
positions than at the leftmost ones. Compared to the standard Gaussian slit widths, the super Gaussian
slit widths show a much better agreement with measured slit widths; the average difference of slit widths
between measured and super (standard) Gaussian functions is ~ 0.01 (0.05) nm. In Fig. 3, an example
of the derived slit functions and fitted preflight slit functions shows that the shapes are very similar.
The wavelength calibrations using different slit functions are characterized for the ozone fitting
window and are shown in Fig. 4b. The shift parameter is determined from irradiance and radiance at
second cross-correlation step after slit parameters are determined from irradiances at first cross-
correlation step. Note that the wavelength shifts fitted between first and second steps are very similar,
indicating little correlation between slit and wavelength calibration parameters. This analysis indicates
that the accuracy of wavelength registration in ozone fitting wavelengths is 0.03-0.06 nm for earthshine
measurements and < 0.02 nm for solar measurements with consistent variation over all cross-track
pixels. These wavelength errors are larger than those reported by Seftor et al. (2014), due to different
fitting windows. They use 350-380 nm where prominent solar Fraunhofer absorption lines exist and the
interference with ozone absorption lines are negligible. Furthermore, the wavelength calibration results
using OMPS measured slit functions show different characteristics from those using both Gaussian-
type slit functions, especially over left cross-track pixels. The different wavelength shifts are likely
because the original OMPS slit functions show slight asymmetry and are used in the wavelength
calibration of L1B data. There exists a ~ 0.07 nm shift between irradiances and radiance. In ozone
retrieval algorithm we shift neither radiance nor irradiance to a reference spectra before retrievals, but
the shift between irradiance and radiance is adjusted during ozone retrievals to account for the on-orbit
variations of wavelength shifts as mentioned in Sect. 2.3.
The right columns of Fig. 2 compare the impact of different slit functions on spectral fitting residuals
of solar irradiances, together with the average fitting residuals as a function of cross-track position in
Fig.4.a. Measured solar spectra are mostly within an average of ~ 1% of modeled solar spectra, except
for the first few wavelengths. Based on these fitting results, we revise the fitting window to 302.5-340
nm. The fitting residuals using a derived standard Gaussian function are the worst for all cross-track
positions. On the other hand, the super Gaussian slit function similarly represents the measured slit
function, but slightly improves the fitting accuracy at the 6~18 cross-track positions (Fig. 4.a). However,
the benefit of using the super Gaussian function for fitting OMPS radiances over the standard Gaussian
function is insignificant within 0.02 % (not shown here). These results agree well with Beirle et al.
(2017), who demonstrated the similar benefit of using Standard and Super Gaussian slit functions on
OMI and GOME-2 measurements. Moreover, the impact of using different slit functions could be less
important for OMPS than OMI and GOME-2 due to its coarser spectral resolution.
In summary, super Gaussian functions are recommended for the OMPS instrument slit functions
than the standard Gaussian functions if the on-orbit instrument slit functions largely deviate from the
preflight-measured slit functions due to instrument degradation or thermal-induced variation. In the rest
of this paper, the measured slit function is used for the analysis of OMPS measurements.

## 3.2 Soft Calibration

The OMPS instrument 2-D CCD detector array could be susceptible to artificial cross-track
dependent errors that are commonly seen in OMI trace gas retrievals. To eliminate this impact on the
OMI L2 product, soft calibration and post-processing cross-track smoothing have been typically
implemented: the first correction removes the systematic wavelength and cross-track dependent
component in measured radiances (Liu et al., 2010; Cai et al., 2012), whereas the second correction
removes cross-track dependent biases in retrievals (Kurosu et al., 2004; Hormann et al., 2016). Figure
5 compares our preliminary tropospheric and stratospheric ozone column retrievals with OMI retrievals
on 14 March 2013. OMPS stratospheric retrievals show an excellent consistency with OMI even though
OMPS measurements does not cover much of the Hartley ozone absorption wavelengths where most of

the vertical information of stratospheric ozone comes from. This is because the separation of stratospheric ozone columns from tropospheric ozone columns is still mainly determined from wavelengths longer than 300 nm (Bak et al., 2013a). On the other hand, tropospheric ozone retrievals are positively biased with respect to OMI, by amounts largely dependent on the OMI cross-track position. Therefore, we decide to include a soft-calibration correction in our retrievals to eliminate wavelength and cross-track dependent errors in OMPS radiances. A general approach to the soft calibration is to characterize systematic differences between measured and computed radiances for scenes where we could assume that all parameters are known; the tropics were typically selected since ozone variability is relatively small (Liu et al., 2010). OMPS normalized radiances are simulated with collocated OMI ozone profiles averaged and interpolated onto $5^{\circ} \times 5^{\circ}$ grid cells to fill in bad pixels mostly caused by the row anomaly. Other forward model inputs are described in Sect. 2. We use 25 days of data between 1 March 2013 and 25 March 2013 under the following conditions: latitude <15$^{\circ}$N/S, solar zenith angle (SZA) < 40$^{\circ}$, cloud fraction < 0.1, and surface reflectivity < 0.1. The systematic and random components of measured-to-simulated radiance ratios are displayed in Fig. 6. Agreement is mostly at the ±2 % level below 310 nm, except at wavelengths shorter than ~ 302.5 nm where the systematic biases increase sharply due to the overcorrection of straylight in OMPS v2.0 data processing. For wavelengths longer than 310 nm, OMPS observations show negative biases with maximum of ~ 3% at 315 nm. The standard deviations of mean differences steadily increase from longer wavelengths to 302.5 nm (2-2.5%) and then sharply rise up to ~ 4 %. The abnormal features of fitting residuals below 302.5 nm shown in Figs. 2 and 6 provide a basis for why we select the lower boundary of the ozone fitting window as 302.5 nm. The soft calibration is applied before the fitting starts by dividing OMPS radiances by the derived correction spectrum just at the initial iteration with the assumption that the systematic biases consistently exist independent of space and time. Figure 7 shows how our tropospheric ozone retrievals are improved with our soft calibration in comparison with retrievals shown in Fig. 5.b. The usefulness of our soft calibration implementation is also evaluated through comparisons of the accuracies of the spectral fitting residuals with and without soft calibration as shown in Fig. 8. The mean fitting residuals without soft calibration are ~ ±1% at shorter wavelengths < 320 nm for all latitudes and sky conditions, whereas for longer wavelengths they increase from 0.3 % to 0.5 % with increasing latitudes. Our soft calibration dramatically improves the fitting accuracy for both clear and cloudy pixels, especially over the tropics and mid-latitude regions; fitting residuals are mostly within 0.2 % at longer wavelengths > 310 nm. In high latitudes, improvements can be identified, but large remaining systematic biases can still be found.

## 3.3 Common Mode Correction

In previous section, it is shown that our soft calibration effectively eliminates systematic biases of measurements relative to VLIDORT simulations for most cases, except for high latitudes/SZAs where there still exists a distinct wavelength-dependent pattern in fitting residuals because the soft calibration spectrum is derived only under small SZA conditions. In order to verify and correct such systematic biases remaining after soft calibration, we characterize spectral fitting residuals at the final iteration classified into 3 latitude/SZA regimes (southern polar region/SZA>60°, tropical region/ SZA<40°, northern polar region/ SZA>60°) for each cross-track position and for one day (14[th] or 15[th]) of each month. The remainder is called the common residual spectrum. Examples of derived common spectra are presented in Fig. 9 for March and August 2013. The main peak positions of residuals of all common residual spectra are well matched to each other. The amplitude of tropical residuals is very similar between two months, whereas the variation of the amplitude at high latitudes seems to be associated with snow/ice cover and SZA variations such that the amplitude is maximized during the polar winter season. Applying the common mode correction (CMC) means subtracting the common spectrum with amplitude determined iteratively along with the rest of state vector components from the measured spectrum. Fig. 10 compares the fitting residuals at high SZAs for one orbit of data on 02 March 2013 with and without the common mode correction. It is evident that wavelength dependent fitting residuals are greatly reduced even for the first few wavelengths, with amplitude of spectral residuals reduced from ∼ 1 % to 0.5 %. Moreover, the common mode correction slightly reduces the standard deviations of residuals. The improvement is seen everywhere as shown in Fig. 11 where RMS of relative fitting residuals (ratio of fitting residuals to measurements error) is displayed for all individual pixels within one orbit.

## 3.4 Measurement Error Correction

The measurement error covariance matrix $S_y$ is one of the essential inputs in an OE based algorithm, because it significantly affects the stability of retrievals and retrieval sensitivities. OMPS L1B v2.0 data contain the relative errors of radiance measurements, but these measurement errors (∼ 0.04 % @ 320 nm) were too small to regularize our ozone fitting process so that many retrievals fail due to negative or large positive ozone values as a result of over fitting. Ideally, the measurement errors need to include not only photon shot noise but also other kinds of random noise errors caused by readout, straylight, dark current, geophysical pseudo-random noise errors due to sub-pixel variability and motion when

taking a measurement, forward model parameter error (random part), and other unknown errors. However, OMPS measurement errors reported in the L1B only include photon shot noise and read-out errors, which underestimate the overall measurement error. For this reason, OMI noise floor (NF) errors instead of OMPS random-noise errors are imposed on our preliminary retrievals, as mentioned in Sect 2.3. However, better signal-to-noise ratios (SNRs) could be expected for OMPS than OMI due to OMPS's coarser spectral and spatial resolutions, as shown from the improved detection limit of OMPS $H_2CO$ retrievals compared to OMI as discussed in Gonzalez Abad et al. (2016). Fig. 11 also implies that there is room for increasing the Degrees of Freedom for Signals (DFS) to current ozone retrievals by regularizing them using the improved measurement error instead of using OMI NF; the ideal value of RMS is one, but our RMS is mostly within 0.4 at low and mid-latitudes. The random-noise component of measurements could be derived from standard deviations of spectral fitting residuals (Cai et al., 2012; Liu et al. 2015b). Fig. 12 shows how we derive the measurement errors to improve our retrievals. We first characterize the minimum measurement errors from fitting residuals under nearly clear-sky condition at SZAs < 40° and cross-track pixels between 4 and 33; note that no radiometric calibration is applied to these fitting residuals. The standard deviations of fitting residuals are nearly invariant at longer wavelengths > 310 nm and show a significant increase from ~ 0.1 % at 310 nm to ~ 0.3 % at 302 nm as plotted with the red dashed line in Fig. 12.a. We eliminate the low-frequency portion of the noises with a 4$^{th}$ order polynomial fit to define the minimum OMPS NF errors as plotted with the red solid line in Fig. 12.a. The derived NF errors are ~ 2 (1.5-4) times smaller than OMI NF errors above (below) 310 nm and thereby could increase the measurement information in our retrievals. We impose the minimum NF errors as a measurement constraint in our algorithm when SZAs are smaller than ~ 20°, whereas they are multiplied by a SNR scaling factor to increase measurement errors as a function of SZAs. Figure 12.b shows an example of how derived measurement errors increase with SZA at the boundary wavelengths of the ozone fitting window, with errors from 0.24 % to 0.45 % for 302.5 nm and from 0.097 % to 0.19 % for 340 nm.

Figure 13 shows the effect of using the derived NF errors on our retrievals. The RMS of fitting residuals increases from 0.2-0.4 to 0.4-0.8 in swath lines 50-350, where SZAs are within ~ 60°, due to SNR increases, whereas the average fitting residuals slightly improves by 0.015 %. Using the new NF errors slightly increases the number of iterations; one or more iterations are required for ~ 24 % of the total retrieved pixels and hence our fitting process converges mostly within 3-4 times, except for thick clouds where the number of iterations increases to 6. Using the derived NF errors significantly increases the retrieval information content. Both stratospheric and tropospheric DFSs are improved by 0.2-0.4 under mild SZAs and by up to 0.2 under high SZAs as shown in Fig. 14, so that tropospheric ozone

retrievals demonstrate ~ 1 DFS in low/mid latitudes, which is similar to OMI retrievals (Liu et al.,
2010a). Fig 15.a shows the retrieved tropospheric ozone column distribution with two radiometric
calibrations (soft, CMC) and OMPS NF errors. Compared to Fig 7.b without CMC and OMI NF errors,
the cross-track dependent noises over the polar region are smoothed due to CMC and the columns are
enhanced in the tropics and the northern mid-latitudes due to OMPS NF errors. Successful tropospheric
retrievals typically require better than 0.2-0.3 % fitting accuracy between measured and modeled
radiances in the Huggins band (310-340 nm) (Munro et al., 1998). Our fitting algorithm meets this
requirement after carefully applying empirical calibrations as shown in Fig 15.b; the average fitting
residuals are within 0.1 % for moderate SZAs, with insignificant dependence on cross-track position.

## 4. Conclusions

The OMI ozone profile algorithm has been adapted and modified to retrieve tropospheric ozone and
ozone profiles from OMPS-NM L1B 2.0 product. To verify the best knowledge of OMPS instrument
slit functions, we evaluate OMPS preflight measured slit functions and analytical slit functions
assuming standard and super Gaussian distributions through cross-correlation using a high-resolution
solar reference spectrum. We also adjust preflight measured slit functions to post-launch OMPS
measurements by broadening/squeezing them by up to 4%, which slightly improves the fitting residuals
at nadir cross-track pixels, but by up to 0.18% (e.g., from 0.75% to 0.6% at the first cross-track position)
at edge pixels. The super Gaussian slit functions better represent OMPS irradiances than the standard
Gaussian and even the preflight measured slit functions, but the fitting residuals of radiances with
different slit functions show insignificant differences. OMPS measured slit functions are finally
implemented in our OMPS ozone fitting retrievals because they take account of the slight dependence
of slit functions on wavelengths.
We perform two kinds of radiometric calibrations to eliminate the systematic components of fitting
residuals. First, we apply "soft calibration" to OMPS radiance before retrievals. This correction
spectrum is derived as a function of wavelength and cross-track position by averaging the ratio of
measured radiances to simulated radiances using collocated OMI ozone profile retrievals in the tropics
under nearly clear-sky conditions for 25 days of May 2013. Applying soft calibration to OMPS radiance
dramatically improves the spectral fitting residuals, especially under low to moderate SZA. The
amplitude of fitting residuals decreases from 1 % to 0.2 %. Therefore, the significant cross-track striping
pattern shown in preliminary OMPS tropospheric ozone retrievals is mostly eliminated. Second, the
CMC is implemented to compensate fitting residuals uncorrected by soft calibration, especially for high

SZA retrievals. This correction spectrum is derived as functions of wavelength and cross-track position by averaging one day's fitting residuals over the tropics and northern/southern high latitude regions, respectively. The amplitude of the correction spectrum is iteratively and simultaneously adjusted with ozone. It is found that the amplitude of the fitting residuals decreases by a factor of 2 due to the CMC over high latitudes.

Our preliminary algorithm uses OMI NF errors to represent measurement constraints because OMPS L1B random-noise errors are too tight to stabilize retrievals. However, we found that OMI NF errors cannot sufficiently constrain our OMPS retrievals, indicating that there is room to increase the retrieval sensitivity to measurement information by improving measurement constraints. Therefore, we derive the minimum NF error corresponding to standard deviations of spectral fitting residuals over the tropics. The derived minimum NF error is ~ 0.097% in 310-340 nm and increases to ~ 0.24 % at 302.5 nm, which is smaller than OMI error by a factor of 1.5-4 below 310 nm and 2 above. We apply this OMPS NF error at SZAs < ~20° and those multiplied by a SNR scaling factor to take into account the decreasing SNR with increasing SZA at SZAs > ~20°; at SZA = 90° errors becomes 0.45 % at 302.5 nm and 0.19 % at 340 nm. Using OMPS NF errors as a retrieval constraint slightly improves the fitting residuals, by 0.015 % on average, and both stratospheric and tropospheric ozone retrieval sensitivity (DFS increases by 0.2-0.4), but requires 1 or more additional iterations for convergence. In this study, we meet the requirement to achieve successful tropospheric ozone retrievals in terms of DFS (> 1) and fitting residuals (<0.2-0.3 %) with empirical calibrations optimized to OMPS L1B measurements. In future work, we will characterize OMPS ozone profile retrievals, present error analysis, and validate retrievals using a reference dataset, to verify that the quality of OMPS ozone retrievals is adequate for scientific use.

## **Acknowledgements**

We acknowledge the OMI and OMPS science teams for providing their satellite data and Glen Jaross for providing useful comments regarding OMPS level 1B v2.0 data. We thank Alexander Vasilkov for allowing the OMPS cloud product to be used in this study. Research at Pusan National University by J. Bak and J.H. Kim was financially supported by the 2016 Post-Doc. Development Program of Pusan National University. Research at the Smithsonian Astrophysical Observatory by X. Liu, K. Chance, and K. Sun was funded by NASA Aura science team program (NNX14AF16G) and the Smithsonian Institution. K. Yang was funded by NASA Suomi NPP science team program (NNX14AR20A).

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

**Table1. Surface and atmospheric input parameters and cross section data used in forward model calculations.**

| Forward model Parameters | Data Source |
|---|---|
| $O_3$ cross sections | Brion et al. (1993) |
| Ozone Profile[a] | OMI ozone profiles from Liu et al. (2010) |
| Temperature profile, surface/tropopause pressure | Daily National Centers for Environmental Prediction (NCEP) final (FNL) operational global analysis data (http://rda.ucar.edu/datasets/ds083.2/) |
| Surface albedo | OMI surface climatology (Kleipool et al., 2008) |
| Cloud fraction | Derived at 347 nm |
| Cloud-top pressure [b] | OMPS Cloud Optical Centroid Pressures (OCPs) (Vasilkov et al., 2014) |

[a]**OMI ozone profiles retrieved at 48×52 km$^2$ with spatial coadding and then interpolated to 5° × 5° to fill bad pixels.**

[b]**OCPs retrieved from OMPS-NM L1B v1.0 measurements using a rotational Raman scattering cloud algorithm.**








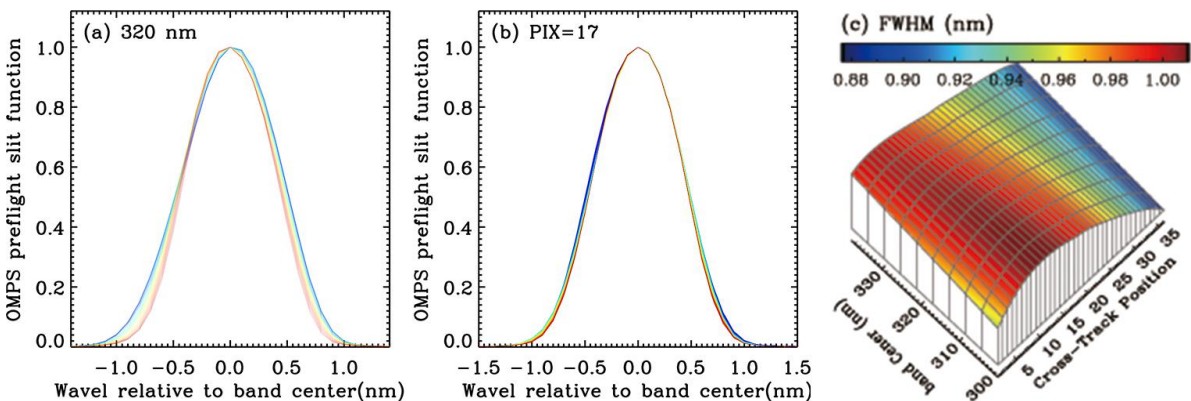


**Figure 1. (a) OMPS preflight slit function at 320 nm band center, with colors representing different cross-**
**track positions from 1 (blue) to 36 (red). (b) Same as (a), but for the 17th cross-track position, with colors**
**representing different wavelengths from 300 nm (blue) to 340 nm (red). (c) Full Width at Half Maximum**
**(FWHM) in nm as functions of cross-track positions (x-axis) and band center wavelengths (y-axis) ranging**
**from 300 to 340 nm.**



















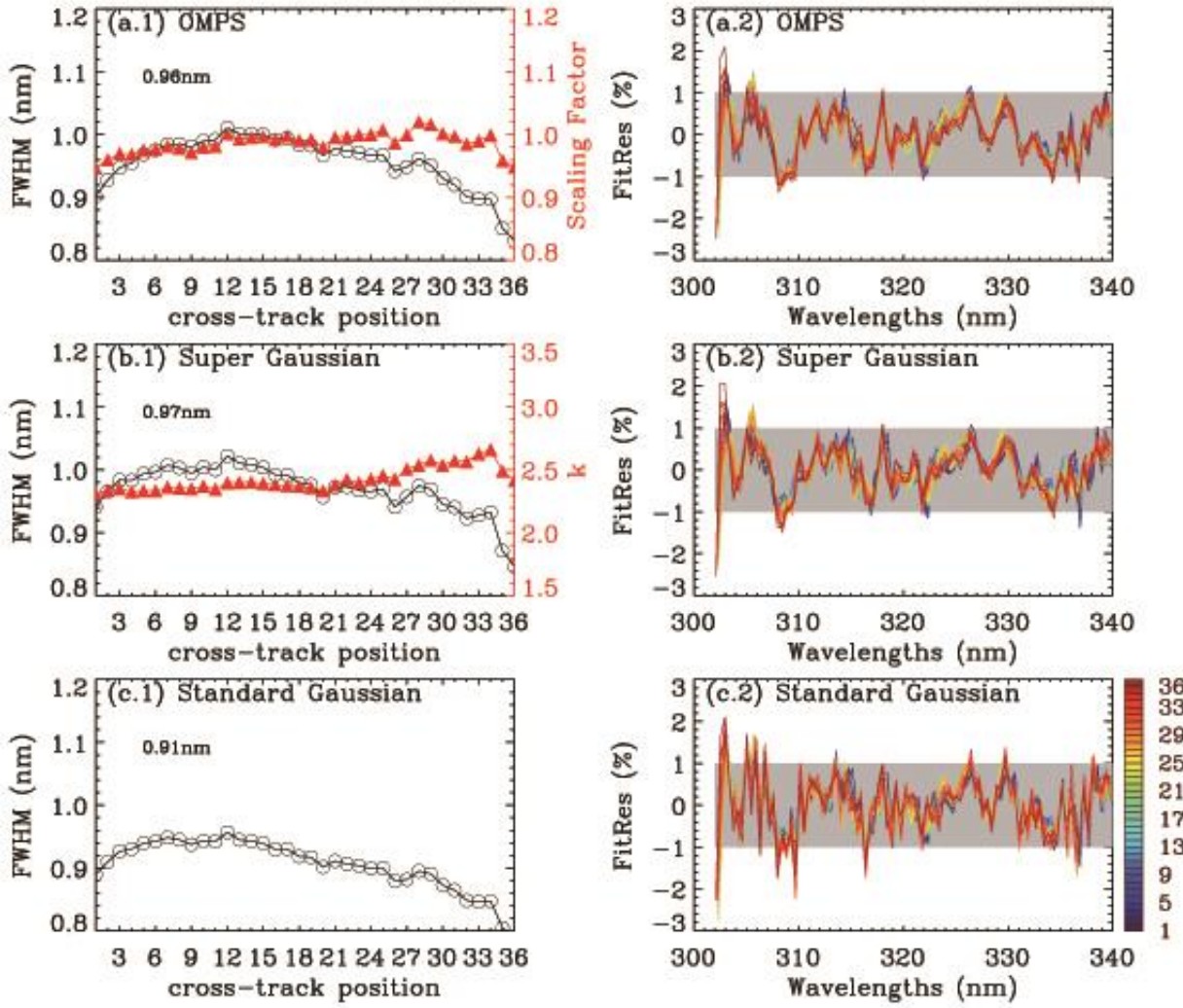


**Figure 2. (Left) Slit function parameters as a function of cross-track position (1[th]-36[th]) for three different slit functions from OMPS irradiance measurements (302-340 nm) for orbit 7132 on 14 March 2013. The legends represent the FWHM averaged over all spectral pixels. (Right) The corresponding relative fitting residuals between measured and simulated irradiance spectra.**








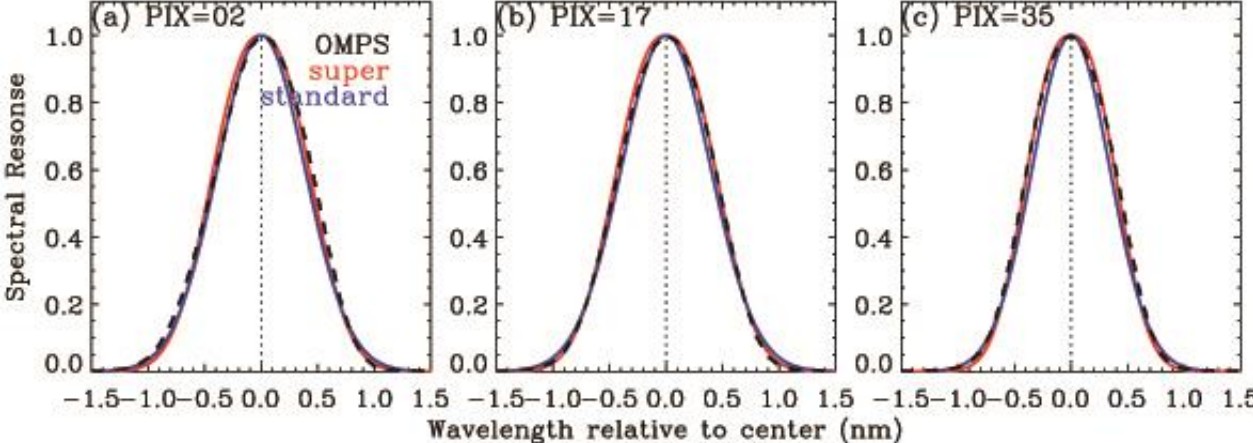

**Figure 3. Comparison of OMPS measured slit measurements (black) and derived slit functions assuming a**
**standard Gaussian (red) and super Gaussian (blue) for orbit 7132.**

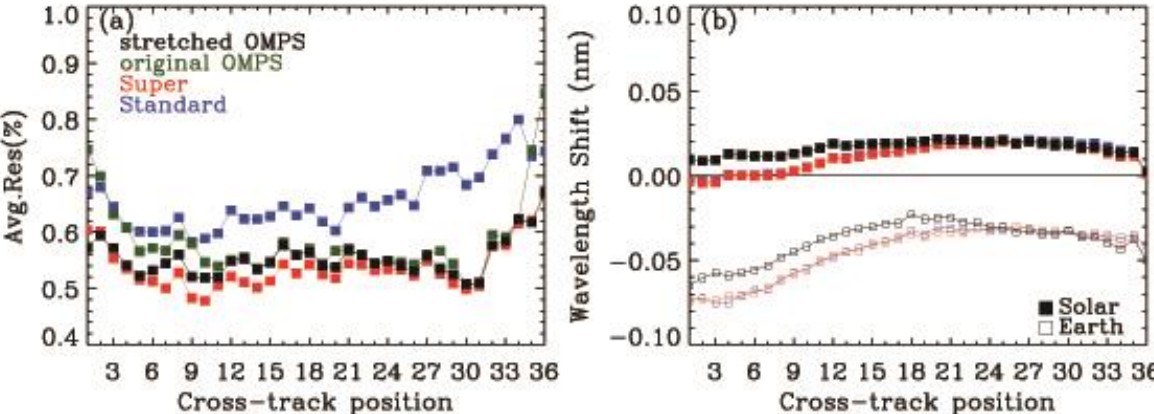


**Figure 4. Same as Fig. 2, but for (a) average fitting residuals (%) as a function of cross-track positions. The**
**green line represents the fitting residuals with measured OMPS slit functions without fitting a scaling factor.**
**(b) Wavelength shifts between OMPS irradiance and reference spectrum (filled symbols) and between**
**OMPS radiance at the middle swath line and reference spectrum (opened symbols).**







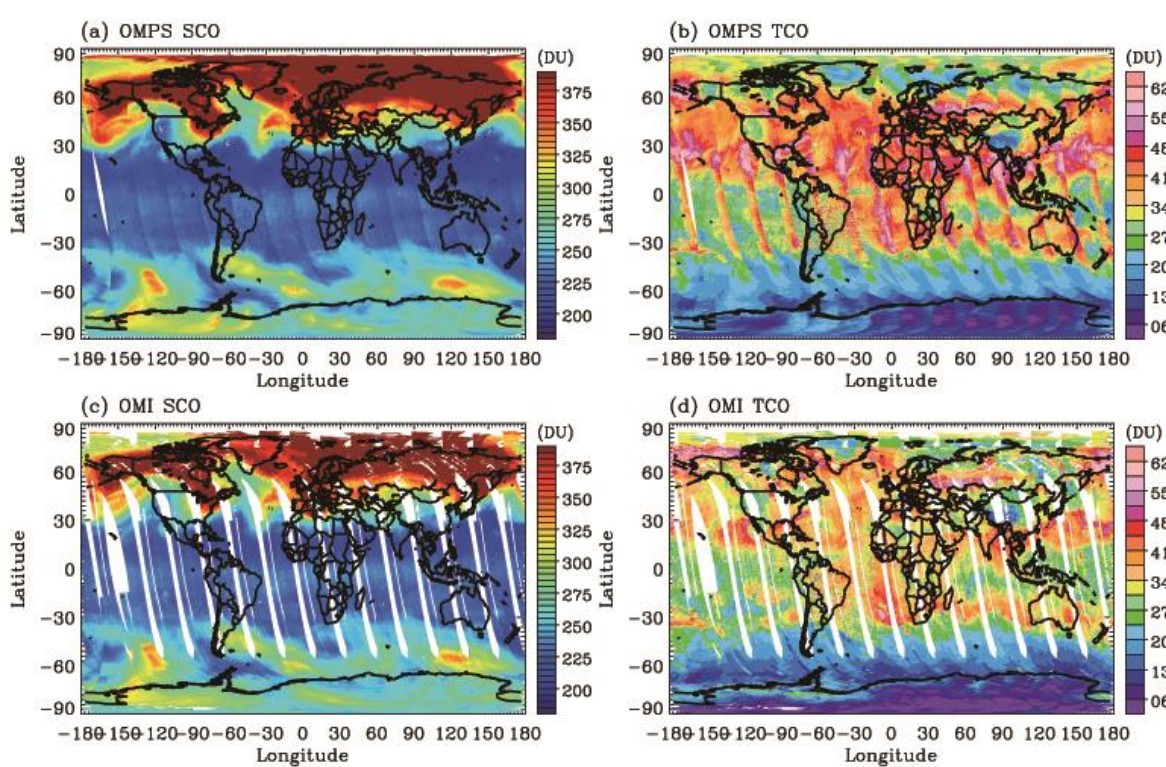


**Figure 5. Maps of stratospheric and tropospheric ozone column on 14 March 2013, retrieved from OMPS (top) without any correction and OMI (bottom) measurements, respectively.**





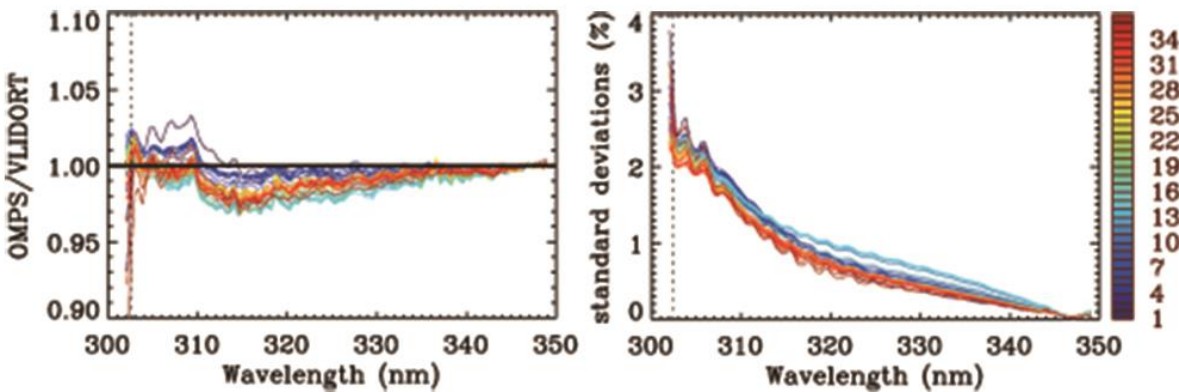


Figure 6. (a) Soft calibration spectrum derived from OMPS measured to simulated radiance ratio at initial iteration, as a function of wavelength ranging from 302 nm to 350 nm. The vertical dotted line indicates 302.5 nm. OMPS data used in this calculation is limited to tropical clear-sky conditions (latitude < $\pm 15°$, cloud fraction < 0.1, surface reflectivity < 0.1) for 25 days between 1 March 2013 and 25 March 2013. Forward model inputs listed in Table 1 are used for OMPS simulations. (b) Standard deviations of fitting residuals. Different colors represent various cross-track positions.

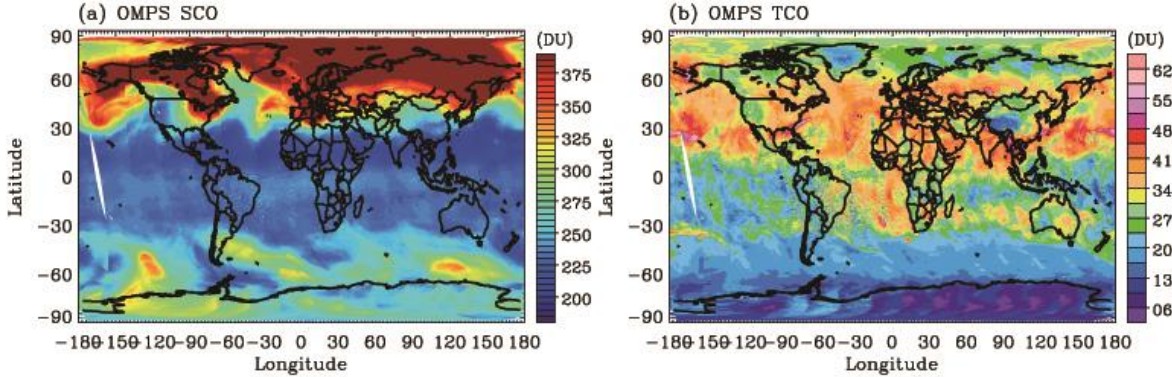


Figure 7. Same as Figure 5 (a) and (b), but for OMPS ozone retrievals with soft calibration.









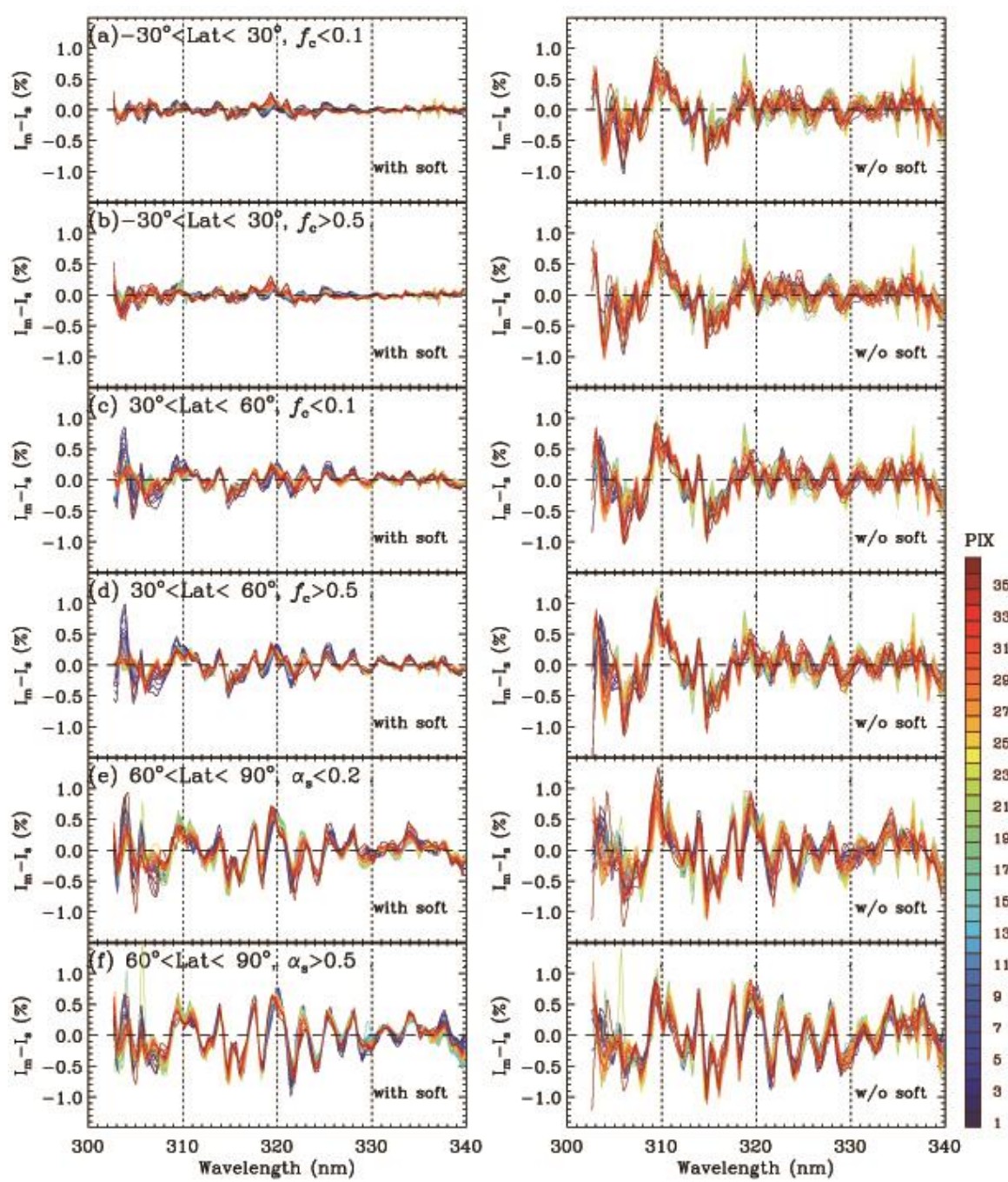


**Figure 8. Comparison of fitting residuals on 14 March 2013 with (left) and without (right) soft calibration for 6 cases: (a-b) Tropics and (c-d) mid-latitudes each for clear sky (effective cloud fraction, fc < 0.1) and cloudy (fc > 0.5) conditions and (e-f) high-latitudes for snow-free and snow-covered surface conditions. Different colors represent different cross-track positions.**

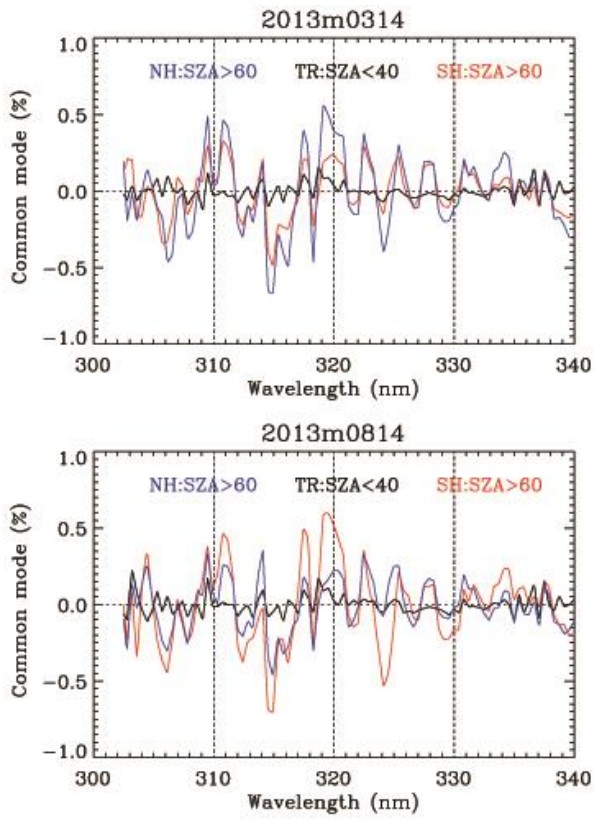

703

Figure 9. Common mode spectra derived from final fitting residuals at the 17th cross-track position using one day of measurements in March (upper) and August (lower), respectively. Note that tropical residuals are derived from nearly clear-sky conditions where SZA < 40°, cloud fraction < 0.1, and surface albedo < 0.1. No special data screening is applied for polar residual spectra, except for SZA > 60°.










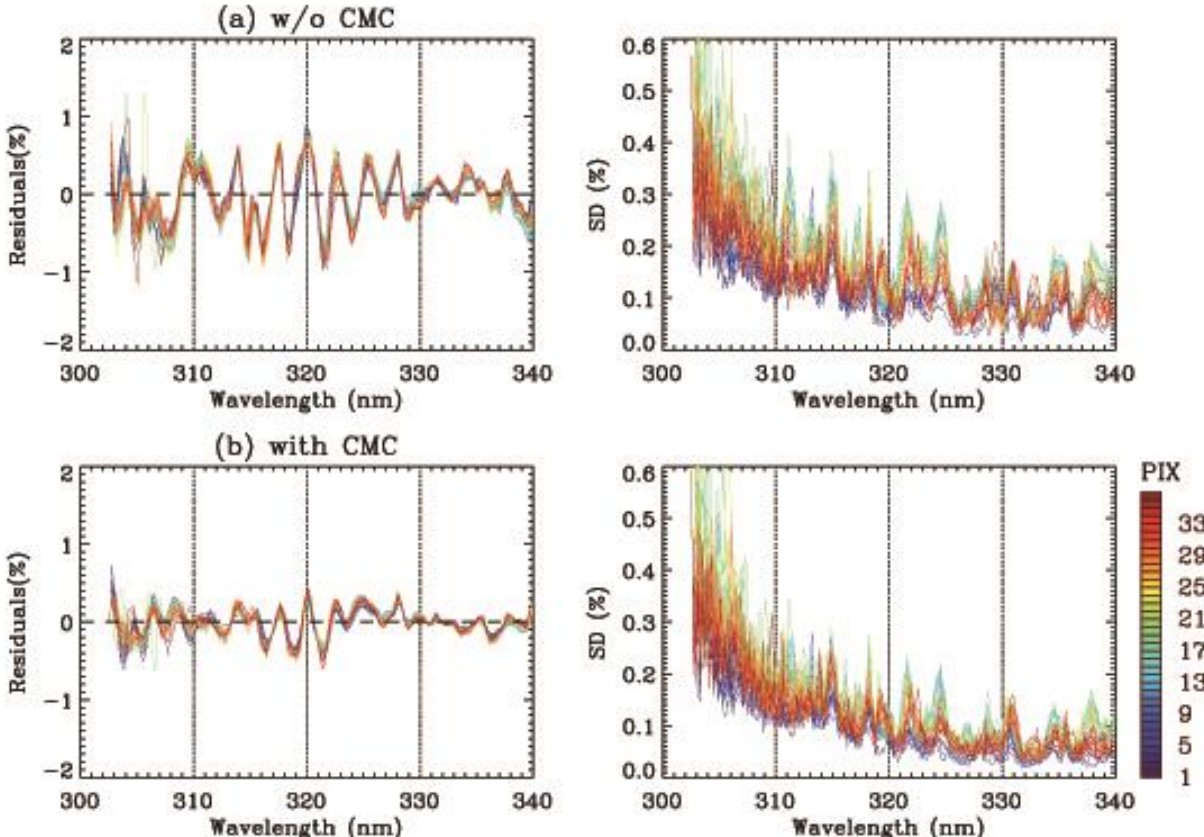

**Figure 10. Comparisons of mean fitting residuals (%) and its standard deviations (%) for latitude > 60°,**
**with different cross-track positions in different colors for one orbit data (6962) on 02 March 2013, without**
**(a) and with (b) common mode correction.**



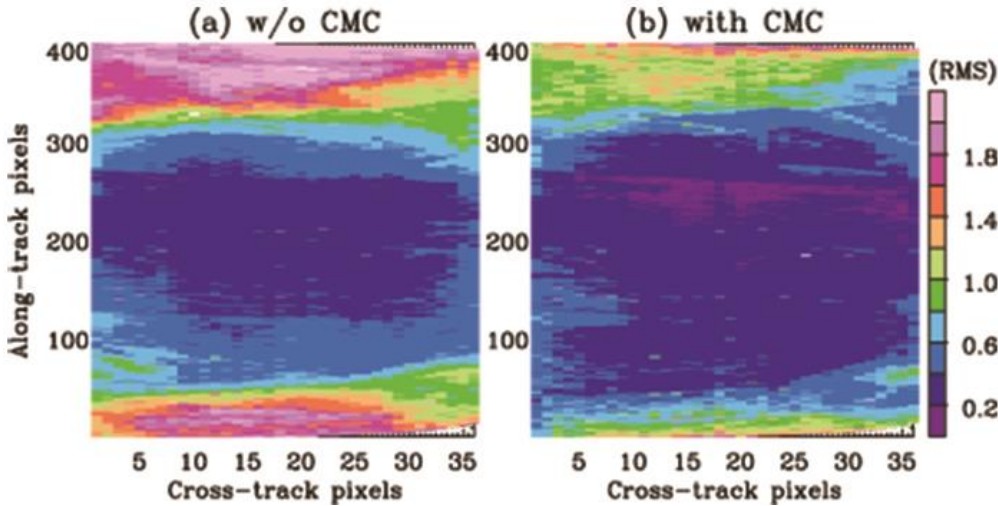


**Figure 11. Same as Figure 10, but for Root Mean Square (RMS) of fitting residuals relative to the measurement errors as functions of along- and cross-track pixels. The RMS is defined as $\sqrt{\frac{1}{n}\sum_{i}^{n}\left(\frac{Y-R}{S_y^{1/2}}\right)^2}$.**

**Note that OMI noise floor errors (0.4% at wavelengths < 310 nm, and 0.2% at wavelengths > 310 nm) are used to define RMS.**

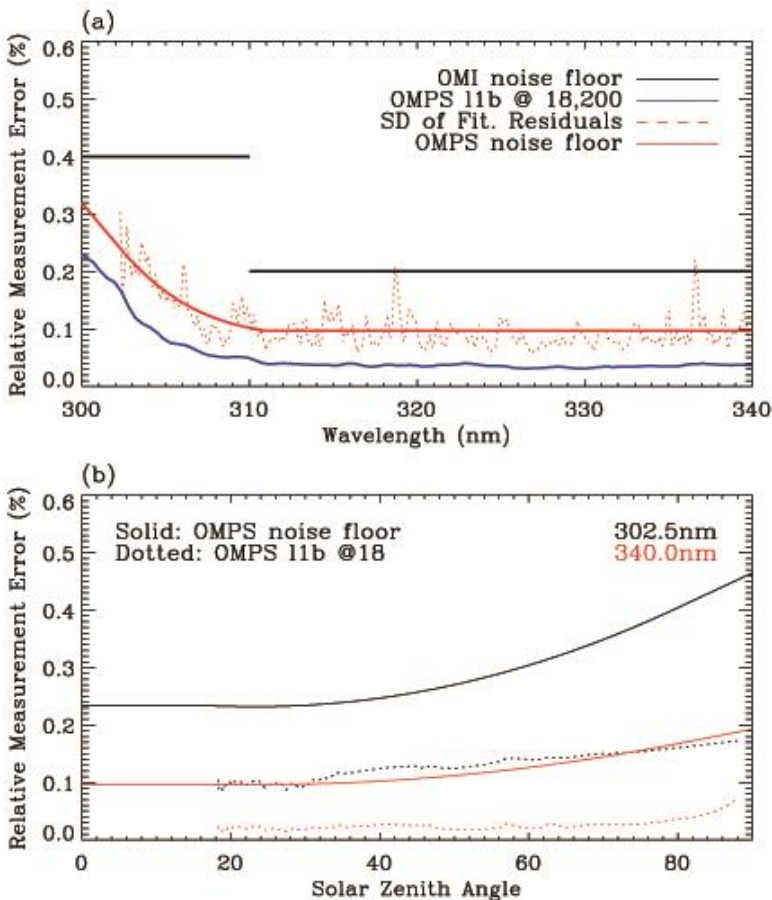

729

**Figure 12. (a) Standard deviations of spectral fitting residuals for 14 March 2013 under clear-sky conditions and for small SZAs < 40° (red dotted line), with the 4th order polynomial fitting of them (red solid line) called "OMPS noise floor (NF) error". This NF error represents the minimum measurement constraint implemented in OMPS ozone fitting process. OMI floor noise error (black line) and OMPS L1B v2.0 random-noise error (blue line) (orbit: 7132, cross-track: 18, along-track: 200) are also shown for comparison in the same panel. (b) OMPS NF at 302.5 nm and 340 nm as a function of SZAs (solid line), with the corresponding OMPS L1B v2.0 measurement error (dotted line).**

737

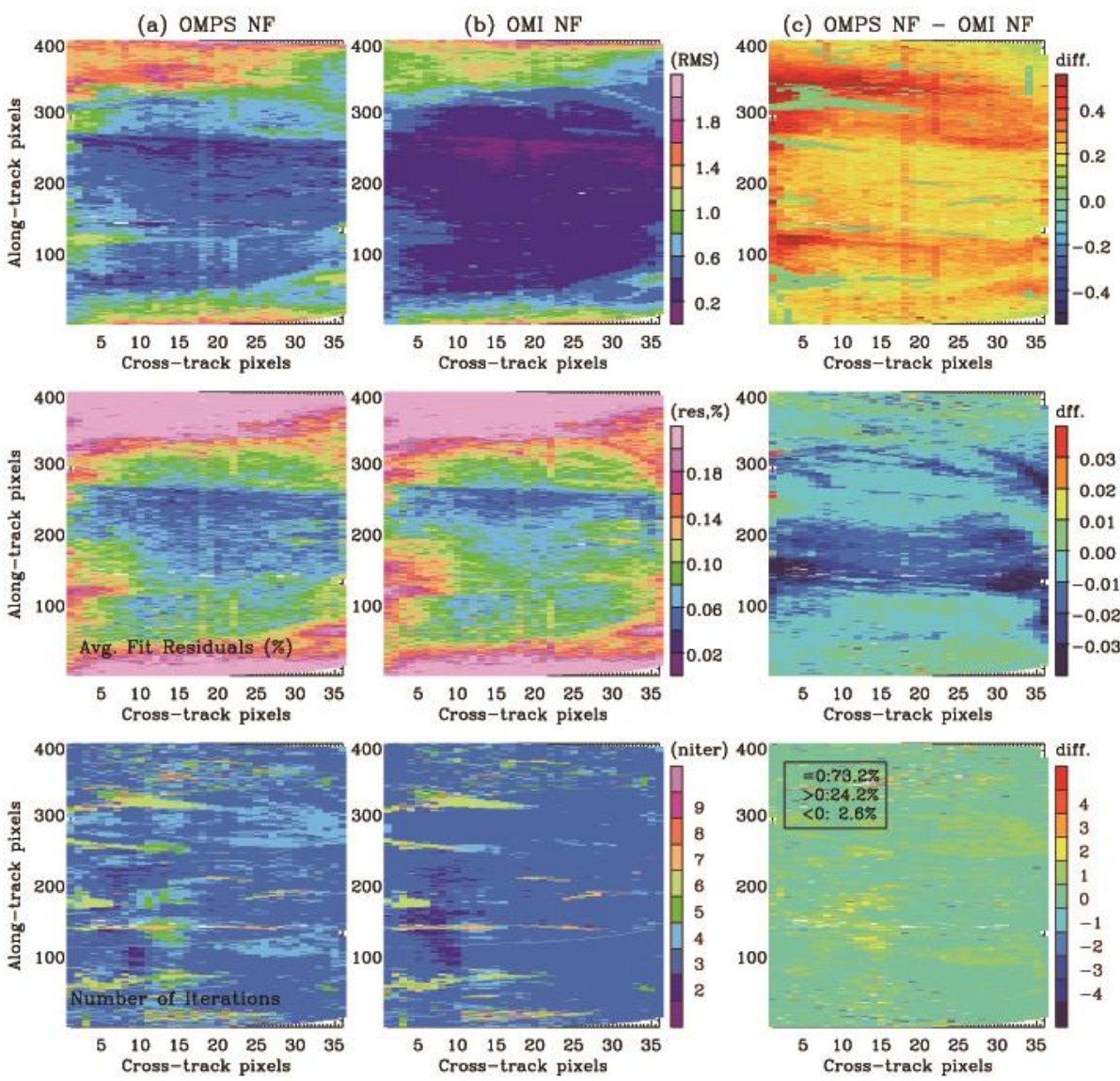

738

**Figure 13. Top: Comparison of RMS of fitting residuals relative to the assumed measurement errors as functions of cross-track and along-track pixels for orbit 7132 with (a) OMPS NF (first column) and (b) OMI NF (second column), respectively, with (c) their absolute differences (third column). The definition of RMS is given in Fig. 11. Middle: Comparison of average fitting residuals relative to the simulated radiances (%), which are similar to RMS, except that radiance differences are normalized to measured radiances instead of measurement errors. Bottom: Comparison of the number of the retrieval iterations.**






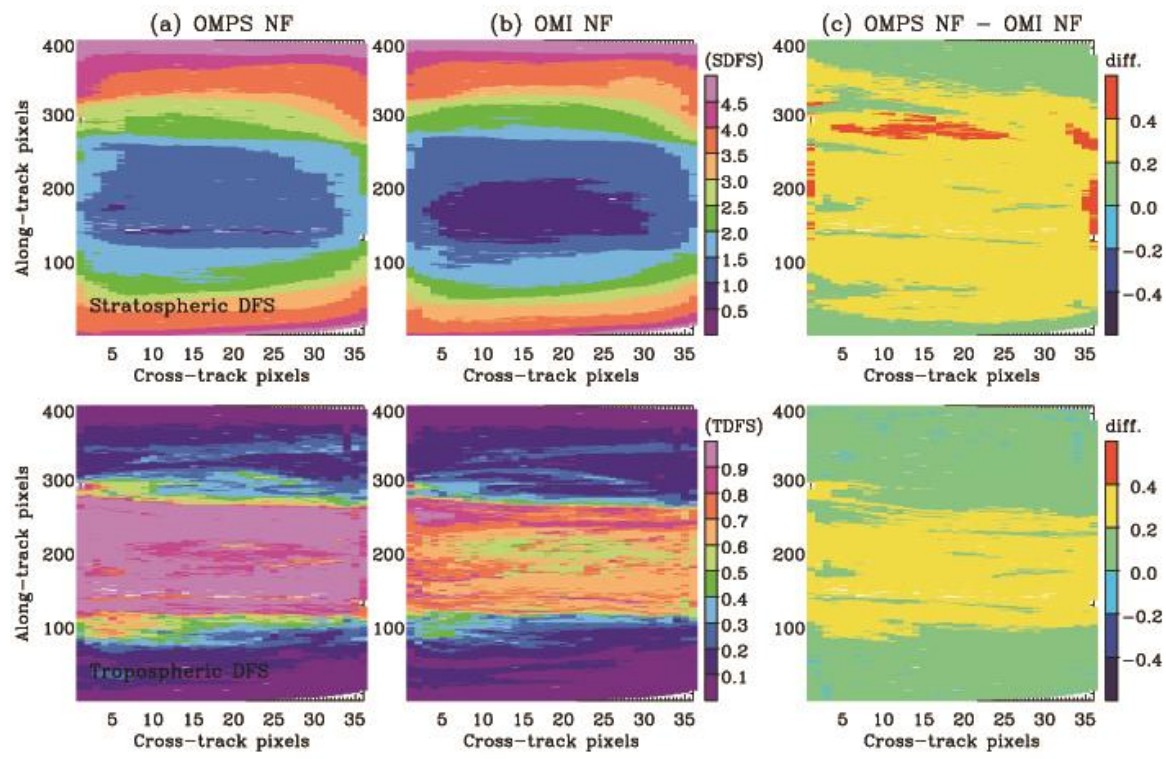


**Figure 14. Same as Fig. 13, but for the integrated Degrees of Freedom for Signal (DFS) in the stratosphere (top) and troposphere (bottom), respectively.**


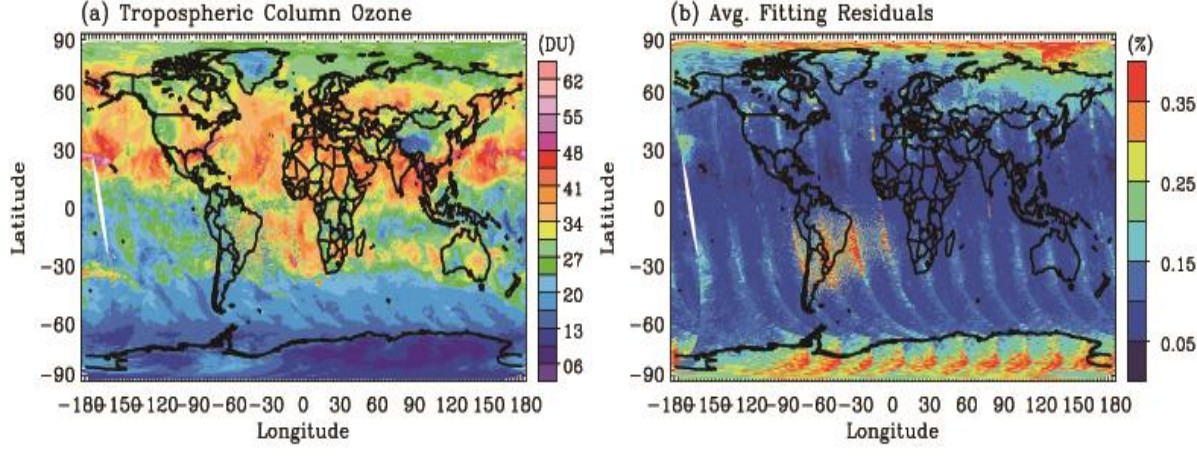


**Figure 15. (a) Same as Fig 7.b, but for improved retrievals with common mode correction and OMPS noise floor error, (b) corresponding average fitting residuals (%).**