# Peer review of "Characterization and Correction of OMPS"

_Atmospheric Measurement Techniques, 2017_

## Referee Comment (RC1) · G. Jaross (Referee) · 28 Sep 2017

I found this paper to be well written an organized, and the scientific relevance clearly indicated. The scientific arguments are substantiated through analysis and presented in a fashion that is mostly understandable. Since this paper reports on a technique that has already been published, its value is in describing how the performance varies with different instruments. The paper accomplishes this objective. It also provides an important independent evaluation of OMPS Level 1 product performance.

I do have several technical questions/criticisms. I don't think they represent major problems, but I would like to see them addressed in some way prior to publication.

Section 3.1

[Figure]

It is not entirely obvious that the discussion in this section is necessary. The authors fail to provide an estimate of their sensitivity to slit function shape that justifies the investigation. Given that they use sun-normalized radiances in their retrievals, much sensitivity to the shape goes away in the ratio. While some sensitivity remains, it is not clear that this represents an error comparable to other error sources. For example, the large OMPS footprint means that most scenes are partially cloudy. The resulting signal gradient across the slit width not only shifts the weighted mean of the function, but also distorts its shape. The effects of this distortion do not cancel in the sun normalization. Surely this is a larger source of error than small shape errors, one that the authors have not accounted for.

Lines 225-226

This is not a correct assumption. OMPS NM is known to have slit widths that change with temperature. The result is Earth-view slit functions that are broader at the swath edges than their irradiance counterparts, by about 4 percent.

Lines 263 and 268

These two lines of text seem contradictory. In the v2 L1B product separate wavelength scales are reported for radiance and irradiance data. These scales differ by at most 0.05 nm. Line 263 implies that the reported radiance band centers are in error by 0.05 nm on average, which is a very large number. But Line 268 states that the derived difference between radiance and irradiance scales is 0.05 nm. Both of these statements cannot be true unless the authors are using the irradiance wavelength scale to represent radiance data. At the very least, the authors should state which parts of the L1B product are in error.

Section 3.3

I fail to understand what is gained from the common mode correction described here. It appears that the end objective is to reduce fitting residuals and standard deviations

along the orbit. When this is done independently for each spectrum, without identifying and correcting the underlying cause of these residuals, it's not clear there are any gains in product accuracy. It would be beneficial if the authors can discuss up front the objectives for these corrections. What types of physical errors will this correction address? Also, I would appreciate a clearer description of how the correction is derived (the explanation in the conclusions is better than in this section).

In addition, I have several editorial comments.

Line 22

. . .resulting in serious. . .

Line 29 (and throughout document)

The typical phrase is "noise floor" rather than "floor noise."

Line 188

The OMPS preflight slit functions were characterized for each CCD pixel . . . (they were not measured for each pixel)

Line 207

"super" instead of "supper"

---

## Referee Comment (RC2) · Anonymous Referee #2 · 13 Oct 2017

General comments and recommendation: This paper demonstrates the need for better retrieving tropospheric ozone from space-borne instrument, OMPS Nadir Mapper. Out of 3 OMPS sensors, the authors explained why OMPS NM is the most suitable for this. More detail description on the retrieval algorithm has been demonstrated on OMI instrument (Liu et al 2010). In this paper, the authors focused on better optimizing OMPS NM L1B measurements by introducing additional tuning methods.

This is a very well-written and well-structure paper and is highly relevant to the community. I only have several technical questions/criticisms.

Specific comments: 1. p2 in the Introduction: Line 53-54: I could only find one reference for Huang et al 2017 in the Reference.

[Figure]

2. Fig 2c.1: It'd be nice to have shown scaling factor on the second Y-axis, as shown in Fig 2a.1 and 2b.1

3. p10, in Soft Calibration Line 296: Figure 5 compares out preliminary tropospheric and stratospheric ozone column retrievals with collocated OMI retrievals... Can you elaborate more on how you collocated OMI and OMPS in this case?

4. p11 in Common Mode Correction You did not have CMC mentioned anywhere in the context, but Figure 10 and 11 use with CMC and without CMC experiments

5. p13 where you have comparison between Fig 15a and Fig 7b. Maybe it is just the color scheme that gives visual illusion. I found the higher value in pinkish color off west Atlantic shown in Fig. 7b, whereas in Fig 15a the max value seems to be shifted to east side of Atlantic in Fig 15a and it seems there is a strong gradient in Fig 15 in the middle of Atlantic.

---

## Author Comment (AC1) · 25 Oct 2017

The author would like to thank for the comments on this manuscript. We replied 5 specific comments from this reviewer.

General comments and recommendation: This paper demonstrates the need for better retrieving tropospheric ozone from space-borne instrument, OMPS Nadir Mapper. Out of 3 OMPS sensors, the authors explained why OMPS NM is the most suitable for this. More detail description on the retrieval algorithm has been demonstrated on OMI instrument (Liu et al 2010). In this paper, the authors focused on better optimizing OMPS NM L1B measurements by introducing additional tuning methods. This is a very well-written and well-structure paper and is highly relevant to the community. I

only have several technical questions/criticisms. Specific comments:

C1. p2 in the Introduction: Line 53-54: I could only find one reference for Huang et al 2017 in the Reference.

A1: We have added another reference, "Huang, G., Liu, X., Chance, K., Yang, K., and Cai, Z.: Validation of 10-year SAO OMI Ozone Profile (PROFOZ) Product Using Aura MLS Measurements, Atmos. Meas. Tech. Discuss., https://doi.org/10.5194/amt-2017-92, in review, 2017b"

C2. Fig 2c.1: It'd be nice to have shown scaling factor on the second Y-axis, as shown in Fig 2a.1 and 2b.1

A2: Super Gaussian function is determined with slit width and shape factor (k), but standard Gaussian function has only one free parameter (slit width) so we don't need to add scaling factor in Fig 2c.1

C3. p10, in Soft Calibration Line 296: Figure 5 compares out preliminary tropospheric and stratospheric ozone column retrievals with collocated OMI retrievals. Can you elaborate more on how you collocated OMI and OMPS in this case?

A3. We did not collocate OMI and OMPS based on pixel by pixel because quantitative comparison is not performed. We compare preliminary OMPS retrievals with OMI on same day (timely collocated). For more clarification, we have removed "collocated" in this sentence.

C4. p11 in Common Mode Correction You did not have CMC mentioned anywhere in the context, but Figure 10 and 11 use with CMC and without CMC experiments.

A4. We have added "CMC" when we first mention Common Mode Correction in section 3.3.

C5. p13 where you have comparison between Fig 15a and Fig 7b. Maybe it is just the color scheme that gives visual illusion. I found the higher value in pinkish color off west

Atlantic shown in Fig. 7b, whereas in Fig 15a the max value seems to be shifted to east side of Atlantic in Fig 15a and it seems there is a strong gradient in Fig 15 in the middle of Atlantic.

A5. Thank for your detailed analysis on this comparison. Calibration and Correction schemes presented by this study clearly demonstrate improvements with respect to spectral fitting accuracy and we will evaluate each scheme with respect to ozone retrieval accuracy using ozonesonde dataset as mentioned in conclusion.

---

## Author Comment (AC2) · 25 Oct 2017

The author would like to thank Dr. Glen Jaross for the constructive and helpful suggestions on this manuscript.
We replied 4 specific comments from this reviewer.

**General Comment**: I found this paper to be well written an organized, and the scientific relevance clearly indicated. The scientific arguments are substantiated through analysis and presented in a fashion that is mostly understandable. Since this paper reports on a technique that has already been published, its value is in describing how the performance varies with different instruments. The paper accomplishes this objective. It also provides an important independent evaluation of OMPS Level1 product performance. I do have several technical questions/criticisms. I don't think they represent major problems, but I would like to see the addressed in some way prior to publication.

**Specific Comment**

C1. Section3.1 it is not entirely obvious that the discussion in this section is necessary. The authors fail to provide an estimate of their sensitivity to slit function shape that justifies the investigation. Given that they use sun-normalized radiances in their retrievals, much sensitivity to the shape goes away in the ratio. While some sensitivity remains, it is not clear that this represents an error comparable to other error sources. For example, the large OMPS foot print means that most scenes are partially cloudy. The resulting signal gradient across the slit width not only shifts the weighted mean of the function, but also distorts its shape. The effects of this distortion do not cancel in the sun normalization. Surely this is a larger source of error than small shape errors, one that the authors have not accounted for.

➔ In this section we would like to figure out if OMPS-NM preflight slit functions are still suitable for representing instrument line shapes after launch for ozone fitting window because it has not been shown in literature, and to determine which analytic function works best to simulate on-orbit instrument line shapes that deviate from the preflight-measured slit functions due to instrument degradation or thermal-induced variation. For this purpose, we justified that super Gaussian slit functions better represent OMPS irradiances than standard Gaussian and even slightly better than preflight measured slit functions. However the fitting accuracy of sun-normalized radiances with different slit functions show insignificant differences due to the differences between the slit functions derived from solar irradiances and slit functions derived from earth radiances caused by scene heterogeneity as also mentioned by this reviewer, differences in stray light between irradiance and radiance. In conclusion OMPS measured slit functions are used in our OMPS ozone fitting retrievals because they take account of the slight wavelength dependence of slit functions. It is worth to know that OMPS measured slit functions work well for ozone fitting window when compared to the use of analytic slit functions.

C2. Lines225-226 this is not a correct assumption. OMPS NM is known to have slit widths that change with temperature. The result is Earth-view slit functions that are broader at the swath edges than their irradiance counter parts, by about 4percent.

We agree that this is not exactly true. But we would like to mention here that in practice, slit functions are typically analytically derived from irradiance spectrum through cross-correlation using high resolution solar reference spectrum and then used to convolve high-resolution cross sections in RT simulation of radiance spectrum if accurate preflight slit function measurements are unavailable. This implicitly assumes that the instrument line shape is similar for both radiance and irradiance. For more clarification, the relevant sentence have been edited as following:

(Before revising) In general, the instrument line shape is assumed to be the same for both radiance and irradiance measurements from satellite observation and determined from irradiances due to lack of atmospheric interference.

(After revising) In general, when accurate measurements of slit functions are not available, the instrument line shape of satellite observation is typically assumed to be the same for both radiance and irradiance measurements, and then can be better determined from irradiances due to lack of atmospheric interference.

C3. Lines 263 and 268 these two lines of text seem contradictory. In the v2 L1B product separate wavelength scales are reported for radiance and irradiance data. These scales differ by almost 0.05 nm. Line263 implies that the reported radiance band centers are in error by 0.05 nm on average, which is a very large number. But Line268 states that the derived difference between radiance and irradiance scales is 0.05 nm. Both of these statements cannot be true unless the authors are using the irradiance wavelength scale to represent radiance data. At the very least, the authors should state which parts of the L1B product are in error.

➔ Yes, the wavelength errors shown in this study are larger than those reported in L1B product, due to the fitting window implemented for the wavelength calibration. OMPS team uses 350-380 nm where prominent solar Fraunhofer absorption lines exist and the interference with ozone absorption lines are negligible, compared to the used spectral region 300-340 nm in this study. As shown in the following figure, the wavelength errors in the 340-380 nm range are reduced to ~ 0.02 nm or less for earthshine measurements and ~ 0.0 nm for solar measurements so that the shifts between solar and earthshine spectra become similar to one reported in Seftor et al. (2014), ~ 0.015 nm at tropics. We have added some sentence in the revised manuscript to notice this fact.

[Figure]

Same as Figure 4.a in the manuscript, but for 340-380 nm.

(Reversed sentence) This analysis indicates that the accuracy of wavelength registration in ozone fitting wavelengths is 0.03-0.06 nm for earthshine measurements and < 0.02 nm for solar measurements with consistent variation over all cross-track pixels. These wavelength errors are larger than those reported by Seftor et al. (2014), due to different fitting windows. They use 350-380 nm where prominent solar Fraunhofer absorption lines exist and the interference with ozone absorption lines are negligible.

C4. Section3.3 I fail to understand what is gained from the common mode correction described here. It appears that the end objective is to reduce fitting residuals and standard deviations along the orbit. When this is done independently for each spectrum, without identifying and correcting the underlying cause of these residuals, it's not clear there are any gains in product accuracy. It would be beneficial if the authors can discuss up front the objectives for these corrections. What types of physical errors will this correction address? Also, I would appreciate a clearer description of how the correction is derived (the explanation in the conclusions is better than in this section).

➔ The soft calibration spectrum is derived from fitting residuals under tropical clear sky condition on March and applied independently on time and spatial. This eliminates fitting residuals very well for mild solar zenith angles, but noticeable systematic biases still remain over high latitude region after soft calibration (Figure 8 left vs right) so that we implement "common mode correction" to correct these fitting residuals, especially for polar region. The idea of common mode correction is to 1) characterize the systematic component of fitting residuals existing after soft calibration as a function of every month and three latitude bands (Southern/Northern high latitude and tropics), which is called "common mode spectrum". 2) The amplitude of this common mode spectrum is adjusted to massage OMI radiances during iterative ozone fitting. Both soft calibration and common mode correction account for errors that could

be not explained by any physical meaning, such as forward model/parameter errors and uncorrected measurement errors. We show benefit of applying "common mode correction "on the fitting accuracy (figs 10 and 11) and on the smoothness of the cross-track dependent noises of the tropospheric ozone retrievals over the polar region (Figs 7b and 15a). As mentioned in conclusion, we will also evaluate algorithm implementation and retrieval accuracy using independent ozone measurements such as ozonesonde and other satellite product.

**Editorial comments.**

Line22 ...resulting in serious...

Line29 (and throughout document) the typical phrase is "noise floor" rather than "floor noise." Line188 The OMPS preflight slit functions were characterized for each CCD pixel... (They were not measured for each pixel)

Line207 "super" instead of "supper"

➔ We have accepted all editorial suggestions. Thanks.